# Transcriptional Activation of a Pro-Inflammatory Response (NF-κB, AP-1, IL-1β) by the *Vibrio cholerae* Cytotoxin (VCC) Monomer through the MAPK Signaling Pathway in the THP-1 Human Macrophage Cell Line

**DOI:** 10.3390/ijms24087272

**Published:** 2023-04-14

**Authors:** Julio Rodrigo Escartín-Gutiérrez, Mariana Ponce-Figueroa, Miguel Ángel Torres-Vega, Leopoldo Aguilar-Faisal, Paula Figueroa-Arredondo

**Affiliations:** 1Doctorate Program in Medical Research, Research Department, Escuela Superior de Medicina, Instituto Politécnico Nacional, Mexico City 07320, Mexico; eskartin@live.com.mx (J.R.E.-G.); jaguilarf@ipn.mx (L.A.-F.); 2Masters in Health Sciences, Postgraduate Studies and Research Section, Escuela Superior de Medicina, Instituto Politécnico Nacional, Mexico City 07320, Mexico; mariana_figueroa@outlook.com; 3Departament of Gastroenterology, Instituto Nacional de Ciencias Médicas y Nutrición Salvador Zubirán, Mexico City 14080, Mexico; miguel.torresv@incmnsz.mx

**Keywords:** *Vibrio cholerae* cytotoxin, VCC, MAPKs, p38, ERK, pro-inflammatory, innate immune response, IL-1β, NF-κB cell death response, AP-1 survival response

## Abstract

This study describes, to some extent, the VCC contribution as an early stimulation of the macrophage lineage. Regarding the onset of the innate immune response caused by infection, the β form of IL-1 is the most important interleukin involved in the onset of the inflammatory innate response. Activated macrophages treated in vitro with VCC induced the activation of the MAPK signaling pathway in a one-hour period, with the activation of transcriptional regulators for a surviving and pro-inflammatory response, suggesting an explanation inspired and supported by the inflammasome physiology. The mechanism of IL-1β production induced by VCC has been gracefully outlined in murine models, using bacterial knockdown mutants and purified molecules; nevertheless, the knowledge of this mechanism in the human immune system is still under study. This work shows the soluble form of 65 kDa of the *Vibrio cholerae* cytotoxin (also known as hemolysin), as it is secreted by the bacteria, inducing the production of IL-1β in the human macrophage cell line THP-1. The mechanism involves triggering the early activation of the signaling pathway MAPKs pERK and p38, with the subsequent activation of (p50) NF-κB and AP-1 (cJun and cFos), determined by real-time quantitation. The evidence shown here supports that the monomeric soluble form of the VCC in the macrophage acts as a modulator of the innate immune response, which is consistent with the assembly of the NLRP3 inflammasome actively releasing IL-1β.

## 1. Introduction

*Vibrio cholerae* is a group of Gram-negative pathogens causing cholerae and other severe diarrheal infectious diseases in humans; the main pathogenic mechanism of these bacteria is the production of toxins. Although the immune cells producing IL-1β are not restricted to macrophages, these are cells in the first line of infection facing pathogens; therefore, the study of the *Vibrio cholerae* cytotoxin (VCC) as a PAMP in the earliest phases of infection is a field of investigation.

The innate immune response is the first line of protection against pathogens, and its components get sensed by pattern recognition receptors (PRR). In turn, these receptors recognize pathogen-associated molecular patterns (PAMPs) for a prompt inflammatory response to begin [1,2]. Eukaryotic cells rely on the family of toll-like receptors (TLRs), which are the innate immune receptors, best characterized in macrophages, since macrophages perform in the first line of immune response (together with dendritic cells). It is known that TLRs stimulation activates signaling pathways, such as mitogen-activated protein kinases (MAPK) and the nuclear transcription factor NF-κB, to stimulate the synthesis and release of pro-inflammatory cytokines. The MAPK signaling controls a wide range of cellular activities of the innate response, and they are particularly important in the regulation of pro-inflammatory cytokine gene expression and programmed cell death [3]. The best characterized MAPK families are: (i) ERK 1 and 2, the extracellular regulated signal protein kinases; (ii) JNK 1, 2 and 3, the c-Jun amino terminal kinases and (iii) MAPK p38, in charge of the onset of the innate immune response [4]. Another important group of PRRs is the Nod-like receptor (NLR) family, which is present primarily in macrophages and dendritic cells and is composed of three domains: the C-terminal region in leucine-rich repeats (LRR), the central nucleotide domain NACHT (NOD) and the N-terminal domain that has either a Caspase recruitment domain (CARD) or a pyrin domain (PYD). The NLRs task is responsible for the constitution of outstanding protein complexes with enzymatic activity denominated inflammasomes, in charge of the launching of the inflammatory process by the further secretion of IL-1β and IL-18. The mechanism used by pore-forming toxins to induce the innate response involves the complementary assembly of the inflammasome, which is a very exciting subject that is still susceptible to further study.

Pore-forming toxins (PFT) such as VCC usually show complex membrane damage mechanisms; they are bacterial soluble monomeric proteins inserting themselves into the lipid bilayer of eukaryotic cells to later assemble five to seven monomers, producing a beta barrel structure responsible for cytoplasmic membrane perforation, resembling and behaving as an active ion permeable channel. Due to its very successful pathogenicity mechanism, PFTs are produced by a large variety of pathogenic bacteria. The toxin pores damaging the cytoplasmic membrane are a cause of profuse diarrhea in patients, also triggering cellular rescue mechanisms that end up acting as subversive processes against the target cell. The *Vibrio cholerae* cytolysin (VCC), formerly known as hemolysin, a very efficiently synthesized PFT, turned out to be a conserved protein efficiently exported by these Gram-negative bacteria. The monomeric VCC is released as a 65 kDa water soluble mature protein; depending on the strain, its production may be very efficient, and in some strains, it is an actively secreted toxin. Once released to the bacterial surroundings, the monomeric form of VCC inserts itself into the cytoplasmic membrane of the target cell due to its transmembrane domain. After the insertion in the cytoplasmic membrane, monomers group one after another, forming a molecular pore of approximately 1–2 nm in diameter [5,6,7], a self-assembled heptameric holotoxin in the shape of a β-barrel structure. Numerous heptameric pores get to disrupt the membrane permeability [8,9]. Each heptameric pore means an injury to the cell when injuries are multiple; eventually, they trigger endocytosis of the damaged membrane, followed by autophagy as a cell-surviving mechanism, which we see at the microscopic level as a striking vacuolating effect. This process of vacuolization involves numerous events of autophagy, eventually leading to apoptotic cell death [10,11].

Although the cytotoxin process of vacuolization has been studied profusely in vitro and in vivo, the innate immune response initiated by the toxin has been usually studied in murine models of infection, using knockout mutants deleting the hlyA gene (encoding the monomeric toxin). In efforts to address, in vitro, the innate response to this cytotoxin, the oligomeric form of the VCC was studied by self-assembly of heptameric pores on liposome membranes and then used to stimulate murine monocytes and macrophages in vitro, showing a pro-inflammatory response [12]. Although all of the above was a good start, testing the monomeric form of the VCC in human macrophages treated in vitro, in our opinion, would be the most direct means to observe the innate response in such a way that it would very probably occur at the early stages of *Vibrio cholerae* infection in patients. Still, the VCC, as a pathogenicity mechanism, is an exciting subject that entertains more possibilities involved in the process of *V. cholerae* bacterial pathogenesis, not only by causing vacuolization or cell lysis but also by starting a pro-inflammatory innate immune response in macrophages. 

In the present study, the soluble form of VCC released by the bacteria was used, and the working concentration was empirically determined in dose-response experiments until a tiny concentration was reached (picograms), which allowed us to avoid cytoplasmic vacuolization but still observe changes due to the pro-inflammatory response (probably by TLRs) and the MAPK activation in response to treatments with the cytotoxin. These changes were determined by performing kinetics of protein activation using Western blots with specific antibodies recognizing MAPK p38 and ERK. The monomeric form of VCC activates the MAPKs pathway in the first hour of interaction with macrophages in vitro. The toxin induces a dual stimulus of initial survival and is then pro-inflammatory, starting the assembly of the inflammasome. 

To better describe the consequences of the MAPK kinases activation, the transcriptional activation of modulators responsible for innate immune response and survival, NF-κB and AP-1, respectively, were determined by quantitative real-time PCR, which revealed the activation of both regulators; this is a striking observation, since AP-1 first conduces to an eventual cell survival program, and the other one, NF-κB, eventually triggers pro-inflammatory cell death (pyroptosis).

## 2. Results

### 2.1. Standardization of the Cytotoxin Working Concentration

In order to know whether the monomer of VCC was capable of activating a pro-inflammatory signaling response, THP-1 macrophages were treated with non-vacuolating concentrations of the VCC, which were empirically determined (40 pg/mL). Figure 1A shows cell morphology changes at times from 0 to 180 min, where the macrophages exhibit changes involving cytoplasmic vacuolization after 60 min of treatment, not earlier. At 90 min, the majority of the cells were vacuolated, demonstrating that the characteristic cytotoxicity can be observed at longer times of exposure. After 180 min of treatment, almost no cells were left, since the treatment caused cell death, consistent with cell shrinking and lysis. In Figure 1B, a limited concentration of the toxin (40 pg/mL) and a longer exposure were standardized to observe pro-IL-1β, explored by western blot. As Figure 1B shows, the non-vacuolating cytotoxicity yields a later activation of pro-IL-1β after 6 h of treatment, with the exception of the negative control (cells without cytotoxin treatment: W/T), suggesting strongly that the soluble VCC monomeric form behaves as a PAMP, with the production of pro-IL-1β. Notice that the cell supernatant does not present the intracellular pro-IL-1β, since it is an immature intracellular form cleaved to assemble as part of the inflammasome platform.

### 2.2. VCC Monomer Induces IL-1β Release

The interleukin IL-1β is the initial pro-inflammatory signal whose production is dependent on the specific activation of the innate immune molecular platform, the inflammasome. In order to know whether the pro-IL-1β expressed is later processed, activated and released, the mature protein IL-1β was determined from the cell supernatant after treatments of the THP-1 macrophages with the standardized limiting concentration of VCC and after evaluating the release of IL-1β on the THP-1 cell supernatant with non-vacuolating concentrations of cytotoxin at 30, 60 and 120 min. LPS was used as the positive control and cells without treatment were used as the negative control. As shown in Figure 2, VCC was capable of releasing IL-1β within 30 min of treatment, and IL-1β increases considerably at 120 min. We next extended the analysis to 2, 4 and 6 h, where IL-1β presents a time-dependent increase. The release of IL-1β with LPS was greater than the one induced by the cytotoxin. As expected, the cells without treatment did not show secretion of IL-1β. These results indicate that the VCC monomer at non-vacuolating concentrations is capable of inducing the release of IL-1β in THP-1 macrophages within 30 min with a time-independent increase.

### 2.3. VCC Activates the p38 Signalling Protein

A MAPK of great interest is p38, specifically participating in the activation of the pro-inflammatory innate immune response, cell differentiation and autophagy of macrophages [13]; consistent with our study, p38 is activated mainly by PAMPs, pro-inflammatory cytokines and endogenous stress inducers.

The results here support that treatments with VCC during 15 and 30 min induced the phosphorylation of p38, in comparison to the rest of the evaluated time periods. The non-phosphorylated form of p38 is constantly present as a baseline in the treated macrophages (Figure 3A). When the relative density of the bands was plotted (Figure 3B), a Gaussian curve peaking at 15 min was evident. In these experiments, the untreated control of THP-1 shows a p38 phosphorylated baseline signal, as it actually happens in the case of phospho-ERK; therefore, only statistically significant activation would be directly related to the treatment with VCC.

In the literature, the activation of p38 has been reported to be related to the loss of intracellular potassium attributed to cell membrane disruption [14,15,16], which is consistent with the pore-forming activity, previously reported by VCC at concentrations in the order of ng/mL and more. This study was designed to avoid the pore formation, which takes place at a toxin concentration that is a thousand times higher; therefore, theoretically under the present conditions (40 pg/mL), the stability of the cell membrane is well kept, so the activation of p38 is actually related to a response originating from receptors triggering the innate immune response.

Consistent with our findings, Porta, in 2002, studying pore-forming toxins, concluded that the activation of p38 will depend on the type of pore-forming toxin and on the target cell, which may result in the triggering of different mechanisms involved in “protection” against PFT (autophagy, vacuolation and, in few cases, in the activation of programmed cell death responses) [15].

### 2.4. VCC Activates the ERK Signaling Protein

Since the activity of the expected MAPK signaling starts in a matter of minutes, in this study, the periods of interaction of VCC with differentiated THP-1 macrophages were studied in lapses in 1 h. Reports from researchers who studied the VCC immune response in mice using *hly*-A knockdown mutants indicate that the MAPK signaling pathway is in fact involved in several aspects of the initiation phase of the toxin’s innate immune response [16]. In turn, the ERK protein is one of the main activation pathways of MAPKs; it holds two known isoforms: ERK 1/2, also referred to as as p44/p42. 

The results from treatments with VCC show higher concentration peaks of ERK1/2 phosphorylation at 15 and 30 min after the addition of the toxin, compared to the incubation periods of 10 and 60 min. Interestingly, there is phosphorylated ERK 1/2 present as a baseline, even without treatment (Figure 4A). The graphic of the relative density (Figure 4B) shows a Gaussian distribution where the highest point of ERK phosphorylation occurs at 15 min of incubation with VCC. The negative control also shows phosphorylated ERK, and it may indicate that there is a pre-existing baseline of the phosphorylated protein concentration, which may not be significant. Colleagues reported previously that ERK phosphorylation is directly involved in the cascade of activation of the transcriptional response of pro-inflammatory proteins such as IL-1β and IL-18. Besides regulation of the immune response, ERK is also involved in cell growth and tissue remodeling [17]; it seems like the main role of ERK is part of a survival program, with the regulation of cell proliferation, differentiation and cell death by apoptosis [18,19]. 

The results shown in Figure 4 suggest that treatments with the monomeric VCC are probably involved in beginning the immunomodulatory activity in THP-1 macrophages, which very probably will start a subsequent assembly and activation of the inflammasome platform.

### 2.5. VCC Monomer Triggers the Activation of AP-1 in THP-1 Macrophages

The Activator Protein-1 (AP-1) transcription factor is linked to the MAPK activation pathway; it is a heterodimeric protein formed by Fos and Jun. To find out whether the VCC monomer was able to activate the AP-1 transcription factor, the mRNA expression of the dimers Fos and Jun was determined in the VCC-treated THP-1 macrophages at the same time periods previously studied.

The RT-PCR results showed that the monomeric VCC induces a rise in the transcription of Fos and Jun genes at 30 and 60 min of treatment, compared to 10 and 15 min or the control without treatment (Figure 5), suggesting that AP-1 is actively functioning.

The transcription factor AP-1 regulates a range of cellular processes such as cell proliferation, differentiation and apoptosis [20]. In turn, JNK is an MAPK present in multiple cell types and has an important role in the innate immune response. JNK is known to activate inflammatory cytokines such as IL-1β [21].

Dr. B. Khilwani [12], in 2015, studied the immune response towards the heptameric VCC; he used artificially assembled heptamers, the oligomeric VCC, assembled into artificial membranes (liposomes). Under those conditions, RAW264.7 and THP-1 cells, he observed the induction of pro-inflammatory responses dependent on the JNK-mediated activation of AP-1, characterized by the nuclear translocation of subunits c-Fos and c-Jun. Experiments from Professor Khilwani are in concordance with the observations here: an increase in the transcription of Jun and Fos at 30 and 60 min, which may be interpreted as the macrophage starting the activation not only of survival signals but also of pro-inflammatory ones [12].

### 2.6. The VCC Monomer Also Triggers the Activation of NF-κB in THP-1 Macrophages

As is well known, the transcription factor NF-κB has an essential role in the expression of pro-inflammatory mediators; it is composed of two subunit heterodimers, p65/p50. As we examined the possibility that the VCC monomer is capable of activating the NF-κB transcription factor, the amount of p50 protein mRNA expressed was measured in THP-1 macrophages treated with the toxin.

The quantitation of the expression of NF-κB (p50) by RT-PCR indicates that THP-1 cells, treated with 40 pg/mL of VCC (Figure 6), show a maximum increase in p50 expression at 60 min. In the positive control stimulated with LPS, the p50 expression is even greater than that in the VCC-stimulated experimental treatment, after the longest incubation of 60 min.

Consistent with these results, signaling pathways that are extrinsically started through p38, ERK and the transcription factor NF-κB, by PAMPs such as LPS, are associated with the further assembly of the inflammasome and the enhanced synthesis of pro-IL-1β [22]. The transcriptional increase in NF-κB-p50 after 60 min observed in these experiments can be interpreted as a VCC-elicited response of the macrophage, starting the innate pro-inflammatory cascade associated with the activation of the multiprotein complex machinery of the inflammasome platform, perhaps already self-ensembled. 

### 2.7. VCC Monomer Activates Caspase-1

IL-1β production can be controlled by the NFκB pathway by activating the platform of the inflammasome mediated by caspase-1. To indirectly determine the presence of the platform of the inflammasome in THP-1 macrophages, the LDH activity was measured in the conditioned media of macrophages previously treated with the VCC monomer at 10, 15, 30 and 60 min; no difference was observed compared to the negative control (Figure 7A). Realizing that this transcription factor begins activation after 60 min (Figure 6), we treated the THP-1 macrophages for 24 h with the VCC monomer and determined the LDH activity (Figure 7A), and in comparison with shorter times of treatment, there is a significant increase in the LDH released, showing cell death after 24 h.

In order to realize whether the observed cell death was actually pyroptosis, the type of cell death caused by the inflammasome, the presence of Caspase-1 was determined at 10, 15, 30 and 60 min. Apparently, at these incubation periods of treatment, Caspase-1 did not have a face-to-face role under the non-vacuolating concentration (results observed in western blots). Therefore, determinations after 6 h of incubation used several controls (LPS, LPS with PMA and VCC + ATP) and the monomer of VCC in non-vacuolating concentrations. The intensity of the bands resulting after treatments with the VCC monomer is more intense than that of the controls with LPS, indicating that during the period of 1 to 6 h, the VCC monomer activates the formation of the inflammasome platform incorporating Caspase-1.

Mamantopoulus studies suggested that El Tor *V. cholerae* hemolysin drives canonical inflammasome-dependent macrophage cell death by western blotting analysis of Caspase-1 and IL-1β [23]. Consistent with our results, the monomer of VCC activates the canonical inflammasome pathway.

## 3. Discussion

Cell receptors starting MAPK signaling after the interaction with PAMPs, such as the toxin monomers, belong to the TLR family. Thus, this study considers that the signaling response is due to a response originating from either TLRs or perhaps the internal receptors NLRs or both, since the TLR binding the monomer might be internalized by the process of external membrane recycling by the endocytic pathway. To be able to demonstrate this interaction in the system, experimental goals had to be switched, so, since the TLR-4/MAPK pathway is active in the THP-1 cell line, we assume that, perhaps, TLR-4 (previously reported in mice) actually initiates the activation of the MAPKs cascade, yielding the experimentally observed cell response.

Then, since TLRs activate the transcription factors NF-κB and AP-1, through MAPKs signaling cascade pathways, VCC actually stimulates one or more TLRs. In the model proposed here, the VCC monomer is detected by a signaling receptor TLR (based on the literature, TLR4 or TLR2/TLR6) [12,24]. In turn, the VCC triggers the activation of the MAPKs p38 and ERK, further triggering the transcriptional activation of Fos/Jun proteins, which are constituents of the heterodimeric transcription factor AP-1. Moreover, VCC also activates the transcription of p50, which, together with p30, bind each other to assemble the heterodimeric transcription factor NF-κB. Both the AP-1 and NF-κB transcription factors are implicated in the inflammatory response; therefore, the monomeric VCC is capable by itself of starting the inflammatory response depending on how long the interaction with membrane receptors is taking place, in response to this particular PAMP.

Colleagues [12,24] previously reported that VCC is sensed by TLR4 or TLR2/TLR6. This work did not address a direct TLR response, instead the signaling pathway MAPKs, the activation of MAPK and the later transcriptional activation of NF-κB and AP-1 in human macrophages. Certainly, further studies are needed to identify whether the recognition receptor may be TLR4 [25], since the signaling is started after intervals shorter than 1 h (as described for LPS responses).

Consistent with this study are the results elegantly obtained by Tetsuya [26], where KO-p38 baby hamster kidney cells were assayed with Aerolysin, the PFT produced by *Aeromonas*, and the cell death rate increased by at least four to five times in comparison to the parental cells [27]. These authors found that part of the transcriptional response towards PFTs is functionally important for defense; it is specifically responsive to PFTs and not the whole bacteria. To our delight, the results from this study concord with those found using the whole bacteria [27]. Two of the transcriptionally induced MAPKs were p38 and JNK; alas, the reason why both p38 and JNK are required to protect the cell in front of PFTs is still under debate, so a proposal mechanism will be offered here.

The results from this work support that the VCC monomeric protein, at very low concentrations of the order of picograms, comparable to the onset of an actual infection, is an early alert to the macrophage and perhaps surrounding cells, starting the signaling from TLRs or NLRs (Figure 8). The treatments with VCC started responses shortly, making it obvious that, at least in macrophages, the most likely response to be stimulated is the one in charge of transcription factor p38, since the innate immune cytokines such as IL-1β production would be the main goal for the first line of protection in charge. As important as the activation of the innate immune response in charge of p38 is, we also wondered whether the cytotoxin also initiates cell death by the pyroptotic pathway. An experimental approach to partially answering the question resulted in the following: cell death led by the toxin was indirectly recognized by the leaking of LDH to the conditioned media, determined after 24 h of treatment (Figure 7A). Cleaved and thus activated Caspase-1 was observed in western blots from the treated cells (Figure 7B).

Moreover, the reasoning suggested by our colleagues describing autophagy [27] as a cell rescue mechanism after a PFT attack now becomes a little undermined. The autophagy and assembly of the inflammasome are perhaps mutually excluding mechanisms regulating cell homeostasis, but the activation of MAPKs is involved in both systems. The concept of the “protection” of the cell towards the injured cell membrane, at least in our study. gets challenged, since results with monomeric VCC show that ERK gets activated perhaps earlier but almost at the same time as p38 (however, within the first hour of incubation, Figure 5). Interaction with various pathogens stimulates the regulation of these pathways in different ways [28]. In this case, the monomer of VCC is a PAMP that, at concentrations of picograms, activates p38 (Figure 2) during the first 15 min of interaction, meaning that the involved mechanisms of the inflammatory response are activated in the first minutes; perhaps, the sensitivity of the macrophages may explain such an efficient response. Additionally, since the kind of mechanism to be activated may depend on the extent of the damage the cell is enduring, it can be the case that the assembly of the inflammasome platform is preferred when the stimulus of the toxin is less severe and the toxin is only added in picograms, rather than the process of autophagy. The process of autophagy may take place only after the cell experiments’ multiple pore formations, such as when concentrations of nanograms are used.

The activation of ERK turns out to be paradoxical, since it is a kinase not directly involved in the innate immune response; nevertheless, it gets activated by treatments with picograms of the monomeric VCC. Activated ERK is indicative of a survival response, since it occurs here under the exposure of a restricted concentration of the monomeric form of a PFT; this fact would only make sense in association with the need for a rather prolonged survival of the cell. Perhaps a prolonged survival response may be needed in a cell destined to undergo the pyroptotic cell death. Before the pyroptotic macrophage becomes necrotic, the cell is actually alive and committed to ensuring an optimal release of pro-inflammatory cytokines such as IL-1β, which are actively produced by the inflammasome and therefore there´s need of an active and ongoing survival response. In other words, the innate immune response program remains in charge of the p38 activated pathway, so the innate cytokines are profusely synthesized by the inflammasome. Once the inflammasome is leading, this response must be kept going as long as possible; therefore, the survival response in charge of ERK (and perhaps JNK) is taking care of ensuring that the inflammasome would continue to be productive until the last minute of the cell viability. In the particular case of pyroptotic macrophages responding to a pathogenic cytotoxin, the ERK pathway relies on keeping the survival signals on before the necrotic cell death strikes (Figure 8).

Because the macrophage is not dead yet; the inflammasome is actively secreting cytokines of the innate response to effectively initiate activation of the second line of defense of the immune system, such as T cells, but very importantly, the T lymphocytes in charge of generating the specific immune response. In vitro studies like these have shown that the monomeric form of VCC is actually a suitable tool for studying cellular functions [29,30,31], such as membrane remodeling and the inhibition of the pore-forming mechanism. With the above reasoning and the results of the present study, we may speculate the following: treated macrophages are setting up a pro-inflammatory response towards the toxin, where an active production of IL-1β is necessary. In order to maintain the most efficient production of IL-1β, the cell must induce a survival response; yet, at the same time, cell death is initiated going forward, as part of the mechanism of the assembly of the inflammasome. A timeline explaining the occurrence of the events is also proposed (Figure 9).

The experimental evidence shown here suggests that the monomeric soluble form of the VCC acts as a modulator of the innate immune response in the THP-1 macrophage system, which is consistent with the assembly of the NLRP3 inflammasome actively releasing IL-1β. Cell death events consistent with pyroptosis were monitored in the treated macrophages by revealing the mature form of Caspase-1, which is consistent with the further release of LDH into the conditioned medium, after 24 h exposure treatments. This study, to some extent, may help in understanding the VCC contribution as an early stimulator of the macrophage; although, until now, it has not shed light onto the lack of long-term protection achieved by engineered *Vibrio cholerae* vaccines.

## 4. Materials and Methods

### 4.1. Chemical Agents and Antibodies

Antibodies against the kinase activated by extracellular receptor (ERK) 1/2, phosphorylated ERK, p38, phosphorylated p38, Caspase-1 and interleukin-1 were all obtained from Cell Signaling (Danvers, MA, USA), and β-actin was obtained from Santa Cruz Biotechnology (Dallas, TX, USA). The kit of reagents for the determination of the release of LDH was obtained from Biovision (Milpitas, CA, USA).

### 4.2. Preparation of the VCC Toxin

The strain of *Vibrio cholerae* 69750 is a super producer of VCC previously reported in [10]. Bacteria were grown in 5 mL of the LB medium at 37 °C, with a constant agitation of 270 rpm overnight. A total of 1 mL of the supernatant was taken from the cells and centrifuged at 14,000× *g* for 1 min in a micro-centrifuge at 4 °C. Subsequently, the supernatant was collected and sterilized by a 0.22 μm membrane filter; finally, the toxin was concentrated by molecular weight exclusion columns (100 kDa and then 50 kDa of Amicon Ultra 0.5 Ultrafiltration system at 14,000× *g* for 10 min); the concentrated toxin was identified by 12% PAGE and then collected and preserved at −4 °C until its use. The biologic effect of the final toxin preparation was evaluated by dose response experiments.

### 4.3. Cell Culture

The human monocyte cell line THP-1 was obtained from ATCC and donated by the Department of Biochemistry of the National Institute of Nutrition and Medical Sciences; it was generously provided by Dr. Sigifredo Pedraza Sánchez. The cells were cultured as recommended: in RPMI 1640 medium supplemented with 2 mM L-glutamine, 10 nM HEPES, 1 mM sodium pyruvate, 0.05 mM β-mercaptoethanol and 10% fetal bovine serum. The cells were incubated at 37 °C in a humidified atmosphere with 5% CO_2_. For the tests, six-well plates were used in which 1.5 × 10^6^ cells per well were used and 50 ng/mL of PMA was added in 2 mL of RPMI 1640 supplemented medium. Subsequently the cells were incubated at 37 °C in a humidified atmosphere with 5% CO_2_ for 48 h, they were exchanged for fresh medium and re-incubated at 37 °C in a humidified atmosphere with 5% CO_2_ overnight and the old medium was removed. A new medium was added, this time using supplemented RPMI, without phenol red and without FBS.

### 4.4. Titration of Biologic Activity (Vacuolating Effect)

A 96-well plate with 1 × 10^6^ Vero cells was seeded with supplemented DMEM and incubated at 37 °C in a humidified atmosphere with 5% CO_2_ overnight; then, the fresh medium was added to the cells without FBS. One volume of the concentrated toxin was added to the first well of the plate; then, 1:2 serial dilutions were made in the following wells. Then, the plate was incubated at 37 °C in a humidified atmosphere with 5% CO_2_ and was observed by inverted optic microscopy every 30 min, keeping aware of the vacuolating effect on the cells, as reported in Figueroa-Arredondo et al., 2001 [10]. The conditions used to standardize the toxin treatments in THP-1-activated phorbol 12-myristate 13-acetate (PMA) macrophages were taken from the Vero cell line results where no vacuolization was found (40 pg/mL) and later adjusted to their protein concentrations (see below). 

The determination of protein concentration of the VCC was performed by the Bradford Method, as described in [32]. The concentration of VCC to be used is determined by its biologic activity; the shorter concentration showing at least 50% of the cells with heavily vacuolated cytoplasm is considered a vacuolating concentration, as established in Figueroa-Arredondo et al., 2001 [10]. In this work, the sub-vacuolating cytotoxin concentration is determined by the lowest concentration and does not produce a vacuolating effect 24 h after the addition of the toxin; instead, LDH is released to the cell culture medium.

### 4.5. Kinetics of the VCC in THP-1 Cells

THP cells that were already differentiated were treated with the sub-vacuolating concentration of VCC standardized as described above, incubating the following periods: 10, 15, 30 and 60 min, respectively. The positive control was 1 µg/mL of LPS; the negative control was cells with no treatment. After incubation, the supernatants were collected and stored at −80 °C, the cells were suspended in 200 μL of RIPA lysis buffer and the plate was scraped with cell lifters. The supernatant was collected and centrifuged at 14,000× *g* for 10 min at 4 °C. The protein concentration was determined by the method (Pierce BCA protein assay kit).

### 4.6. Immunoblot

Collected cells from the treatments described above were kept frozen at −80 °C in lysis buffer added with a cocktail of protein cOmplete Lysis, Roche, Basel, Switzerland). From here 10 μg of the protein from each treatment was loaded in a pre-made denaturing electrophoresis gel of 10% acrylamide (Invitrogen Novex, Carlsbad, CA, USA), electrophoresed for 15 min at 80 V and then electrophoresed for 75 min at 150 V. Electrophoresis-separated protein bands were transferred to a polyvinylidene fluoride membrane (Invitrogen Novex PVDF membrane) and then pre-treated with 5% bovine serum albumin in Tris buffer plus 0.1% Tween 20 at room temperature for 1 h. Blotted membranes were incubated overnight in the primary antibody at 4 °C and then subsequently incubated with the secondary antibody for 1 h at room temperature. Luminol was used in the detection of electrophoretic bands and analyzed on a photo-documenter (ChemiDoc XRS+). The relative densities from the bands were estimated with the ImageJ software (NIH, Bethesda, MD, USA). The antibodies used in this study were mentioned above.

### 4.7. LDH Test

Supernatants from each kinetic assay (25 μL) were taken and added to a 96-well plate, and 25 μL of the working solution was added to each well. The appropriate standard curve of LDH and the working solution were made according to the manufacturer’s instructions (Biovision, Waltham, MA, USA). The optical density was measured at 450 nm (time zero), followed by incubation at 37 °C for 30 min with constant shaking (300 rpm). The optical density (450 nm) was measured after a 30 min period of activity. The LDH activity of every treatment was determined according to the formula provided by the manufacturer. These measurements were performed in duplicate from three different experiments, and the graphics were constructed from the collected data.

### 4.8. qRT-PCR

The sub-vacuolating concentration of VCC was added to 1 × 10^6^ THP-differentiated cells, followed by incubation for 10, 15, 30 and 60 min, respectively. The positive control was 1 mg of LPS, and the negative control was the cells without treatment. Total RNA was extracted using Roche’s High Pure RNA Tissue kit, and then the cDNA from each treatment was synthesized from approximately 500 ng of RNA using Roche’s First Strand cDNA Synthesis Transcription Kit. The generated cDNA was used to perform duplicates of qRT-PCR assays using Roche’s Light Cycler TaqMan Master. The following primers were used: p50_Fwd:5′-caccgaagcaattgaagtga-3′, p50_Rev: 5′-ggcctgagaggtggtcttc-3′, jun_Fwd: 5′-ccaaaggatagtgcgatgttt-3′, jun_Rev: 5′-ctgtccctctccactgcaac-3′, fos_Fwd: 5′-ctaccactcacccgcagact-3′, fos_Rev: 5′-aggtccgtgcagaagtcct-3′, gapdh_Fwd: 5′-agccacatcgctcagacac-3′ and gapdh_Rev: 5′-gcccaatacgaccaaatcc-3′. Glyceraldehyde-3-phosphate dehydrogenase (GAPDH) was used as a housekeeping control gene. All reactions were performed in the LightCycler 2.0 (Roche, Basel, Switzerland) real-time equipment, and the relative transcription results were normalized against GAPDH using the ΔΔCt method [33].

### 4.9. VCC Super Producer V. cholerae 69750 

Purified VCC was obtained from the *V. cholerae* 69750 strain [10] it was cultivated, selected by its natural overexpression of the VCC. Bacterial culture supernatants were taken, filter sterilized (0.22 μm). Since the VCC toxin in its monomeric form has a molecular weight of 65 kDa, molecular weight chromatography was performed by centrifugation, to exclude molecules greater than 100 kDa. Then, another chromatography was performed to exclude molecular weight molecules lower than 50 kDa. In the end the range of molecules kept were between 50 and 100 kDa. The last step was performing PAGE in acrylamide-bis for electroelution of the 65 kDa in 1 mL of PBS and that was the concentrated stock. Aliquots of every step of purification were then electrophoresed as evidence that the protein purification has been carried out successfully. As expected, a band corresponding to 65 kDa was observed in the lane where the concentrate stock was loaded, corresponding to the VCC molecular weight. For corroboration, Purification step samples were electrophoresed then WB, and revealed with an anti-VCC-specific antibody (rabbit polyclonal kindly provided by Professor Takeshi Honda) as it is shown in Appendix A. The 65 kDa protein corresponding to the VCC in the western blot was identified.

### 4.10. Standardization of the Signalling Responsive Concentration of VCC

In order to define the optimum concentration of the VCC to be used in signaling experiments that specifically avoid the vacuolating effect more related to autophagy [34], the standardization of a sub-vacuolating concentration was empirically determined. Initially, the use of the Vero cell line was considered to be convenient, since the vacuolating effect was previously characterized in this cell line [10]. The sub-vacuolating concentration was estimated after serial decimal dilutions of the concentrated stock, considered to be the last dilution where no vacuolating effect was observed after 12 h of incubation. different concentrations were tested, by optical microscopy. Observations were performed after 1 h until 24 h were completed. The typical vacuolating effect was observed at concentrations of 168 pg/mL (Appendix A); however, after serial dilutions concentrations of the order of 40 pg/mL were reached and although not vacuolating effect was observed after 12 h, other cytotoxic changes consistent with lysis were noticed. For example, LDH release was detected from the treated cell supernatants after 24 h; therefore, the concentration of 40 pg/mL was selected, since the goal of this project is to study the toxin concentrations only capable of starting signaling cascades towards the activation of the MAPK pathways.

Previous studies reported the cytotoxin of *V. cholerae* to be able to induce an innate immune response through the ensemble and activation of the inflammasome protein complex [23,35], leading to pyroptosis, the pro-inflammatory cell death [36]. To determine if the monomer VCC was able to induce such type of cell death, the presence of LDH released in the treatment supernatants was determined at 10, 15, 30 and 60 min of treatments with the VCC and after 24 h of treatment (Figure 1B). LDH unequivocally indicates lysis of the treated cells; therefore, pyroptosis is highly probable to occur.

### 4.11. Standardization of the Signalling Responsive Concentration of VCC

For each experiment, data were obtained in triplicate and were reported as the means ± S.D. Comparisons of the means were analyzed using ANOVA plus the Student–Newman–Keuls method, and significant values are represented at *p* < 0.05. The *p* values are indicated as follows: * *p* < 0.001 and ** *p* < 0.05.

## 5. Conclusions

The use of limited non-vacuolating concentrations of VCC, in the order of picograms, was sufficient for the induction of a complex transcriptional response of THP-1 differentiated macrophages. This study presents experimental evidence supporting that the soluble monomeric form of the *V. cholerae* cytolysin VCC elicits a proinflammatory response in human macrophages, activating the p38 MAPK early after exposure with the toxin and the production of pro-IL-1β, followed by NF-κB activation. As usual, NF-κB activates its own transcription for the further amplification of the started pro-inflammatory response. Exposure to the toxin involves not only the transcriptional activation of pro-IL-1β but also the one of Caspase-1, which eventually will render the mature IL-1β by the assembly and activation of the molecular platform of the inflammasome. 

On the other hand, almost synchronically, a cell survival response comes up as early as 10 min after exposure to the toxin, since the survival member of the MAPK pathway, the ERK MAPK, was activated, with further cell survival activity revealed by the activation of the transcription factor AP-1 (Jun and Fos phospholylation). It is necessary to point out that a survival response only makes sense in a cell such as the activated macrophage, destined to undergo the pro-inflammatory lytic cell death known as pyroptosis (suggested strongly by the release of LDH to the culture supernatants). The cell is programmed for the assembly and activation of the inflammasome for actively producing IL-1β to be the innate immune cytokine destinated to recruit the cells that will initiate the secondary, more specific immune response. In this particular case, the activated macrophage, starting and maintaining a program of cell survival, will ensure a maximum release of IL-1β before the cell actually undergoes the classical lysis, characteristic of the proinflammatory cell death named pyroptosis. More studies further supporting these conclusions are definitely needed.

## Figures and Tables

**Figure 1 ijms-24-07272-f001:**
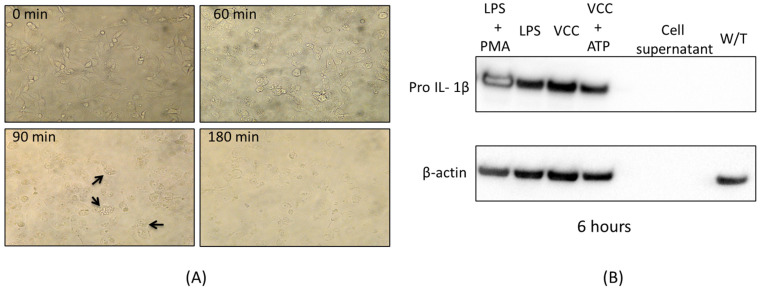
Standardization of responsive concentrations of VCC. A restricted concentration of VCC (40 pg/mL) induces pro-IL-1β after 6 h of treatment without cytoplasmic vacuolization in differentiated THP-1 macrophages. (**A**) Chronological sequence by inverted optical microscopy. The cells were treated with 40 pg/mL toxin and later monitored (0, 60, 90 and 180 min) during incubation and photographed from an inverted optical microscope (40× Olympus IX71). Black arrows indicate a few characteristic vacuoles at 90 min; yet, this is not considered a vacuolating effect (where at least 50% of the cells are vacuolated). The results shown were arbitrarily chosen, but they are representative of three independent experiments. (**B**) Western blot showing Pro IL-1β by treatments with 40 pg/mL of VCC after 6 h of incubation with no vacuolating effect. The expression of pro IL-1β protein with different treatments (LPS + PMA, LPS, VCC, VCC + ATP) at 6 h was measured by a western blot assay. WB of the β-actin protein level is an internal control.

**Figure 2 ijms-24-07272-f002:**
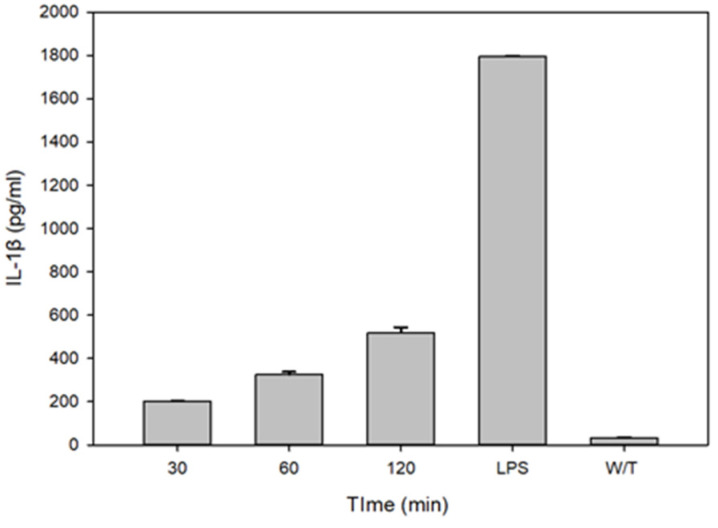
IL-1β release under VCC treatment. THP-1 macrophages were treated with 40 pg/mL of VCC. A positive control of LPS (1 µg/mL) was used. The release of IL-1β was determined by ELISA. ELISA shows IL-1β release under treatments with 40 pg/mL of VCC after 30, 60 and 120 min. Results in the figure are representative of three performed experiments. W/T, without treatment. Mean ± SD. *p* < 0.001, significantly different from the group treated with LPS (ANOVA plus the Student–Newman–Keuls method).

**Figure 3 ijms-24-07272-f003:**
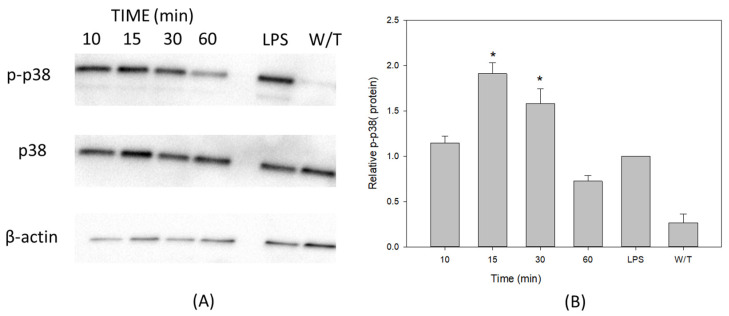
VCC activates the phosphorylation of p38 in THP-1 macrophages. THP-1 macrophages were treated with 40 pg/mL of VCC at 10, 15, 30 and 60 min. A positive control of LPS (1 µg/mL) was used. Cell lysates were analyzed by the Western blot assay. (**A**) Phosphorylation of p38 after treatments with VCC. The results illustrated here are from a single experiment representing three separate experiments. In the upper panel of the Western blot, changes in the phosphorylation of P-p38 were identified, while in the middle panel, it is observed that the content of non-phosphorylated p38 appears constantly in all the analyzed samples. The lower panel shows the β-actin protein considered an internal control. (**B**) Kinetic of treatment with VCC. Results in the graphic are representative of three performed experiments. W/T, without treatment. Mean ± SD. * *p* < 0.001, significantly different from the group treated with LPS (ANOVA plus the Student–Newman–Keuls method).

**Figure 4 ijms-24-07272-f004:**
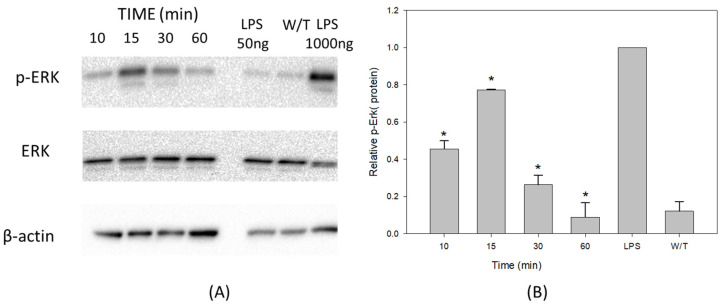
VCC activates the phosphorylation of ERK in THP-1. Differentiated THP-1 macrophages were treated with the VCC at 10, 15, 30 and 60 min. The positive control was LPS (1 µg/mL). Cell lysates were analyzed by Western blot assays. (**A**) Western blot showing ERK phosphorylation under treatments with 40 pg/mL of VCC at 10, 15, 30 and 60 min. The results illustrated are a single experiment representative of three; changes in the phosphorylation of p-ERK are identified in the heading of the Western blot; below this, it is shown that the ERK content is constant in all the samples analyzed. The lower panel shows the β-actin protein at a level considered as an internal control. (**B**) Kinetics of Erk phosphorylation induced by treatments with VCC. The graphic represents results from three experiments. W/T, without treatment. Mean ± SD. * *p* < 0.001, significantly different from the group treated with LPS (ANOVA plus the Student–Newman–Keuls method).

**Figure 5 ijms-24-07272-f005:**
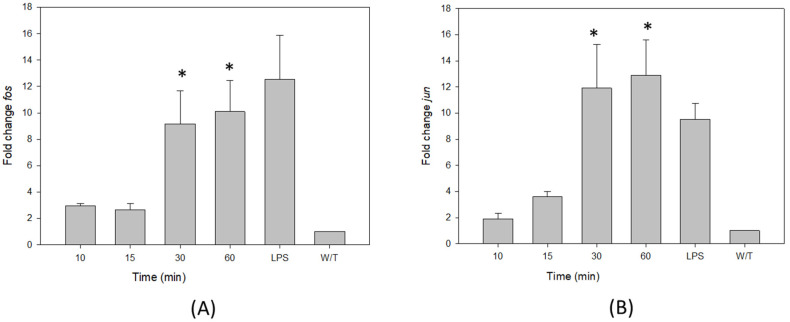
The VCC initiates messenger RNA expression of the transcription factor AP-1. Kinetics of transcriptional activation in THP-1 cells were performed by treatments with 40 pg/mL VCC during 10, 15, 30 and 60 min. LPS (1 µg/mL) was the positive control. (**A**) Transcriptional activation of Fos. VCC starts the transcriptional regulation of the subunits of AP-1 Fos and Jun; then, its mRNA expression was studied by qp-PCR. (**B**) Transcriptional activation of Jun. VCC starts the transcriptional regulation of subunits of Jun, and its mRNA expression was evaluated using qp-PCR. Changes in transcriptional expression were established in comparison with the untreated control and then normalized with the housekeeping gene NAPDH. The graphic includes results from three performed experiments. W/T, without treatment. Mean ± SD. * *p* < 0.001, significantly different from the group without treatment (ANOVA plus the Student–Newman–Keuls method).

**Figure 6 ijms-24-07272-f006:**
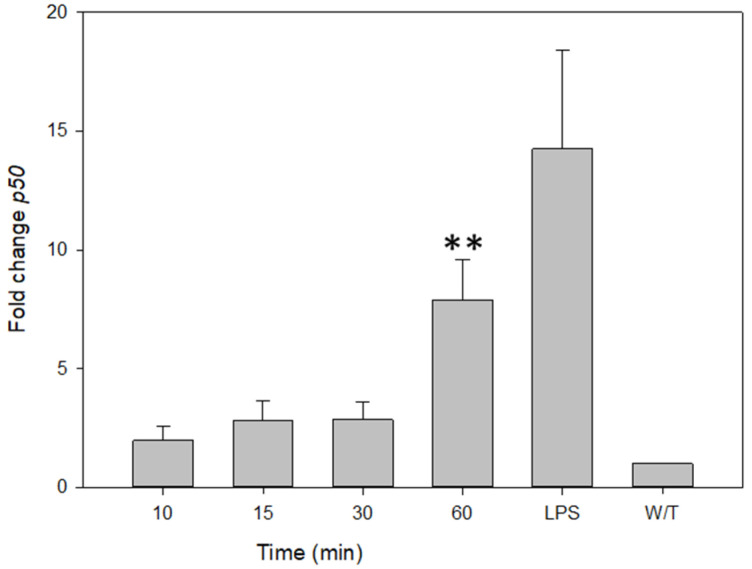
The VCC initiates the expression of the transcription factor NF-κB. THP-1 cells were treated with the VCC toxin at 10, 15, 30 and 60 min. LPS (1 µg/mL) was used as a positive control. The VCC starts the transcriptional regulation of subunits of NF-κB, p50. The mRNA expressions were evaluated using qp-PCR. Changes in expression were compared with the control without treatment and normalized with the housekeeping gene NAPDH. The graphic includes results from three performed experiments. W/T, without treatment. Mean ± SD. ** *p* < 0.05, significantly different from that without treatment (ANOVA plus the Student–Newman–Keuls method).

**Figure 7 ijms-24-07272-f007:**
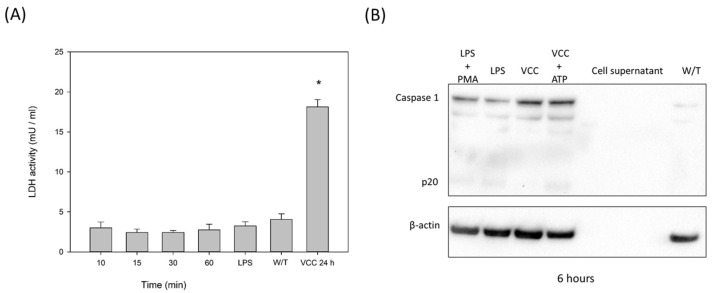
The VCC initiates the expression of the transcription factor NF-κB. (**A**) THP-1 cells were exposed to VCC (0, 10, 15, 30, 60 min and 24 h); the LDH activity was detected and determined from the collected conditioned media. Data are the mean ± SD from three independent experiments. W/T, without treatment. * *p* < 0.001 (ANOVA plus the Student–Newman–Keuls method). (**B**) Expression of Caspase-1 protein with different treatments (LPS and PMA, LPS, VCC, VCC + ATP) at 6 h was measured by a western blot assay. The β-actin protein level is shown as the internal control.

**Figure 8 ijms-24-07272-f008:**
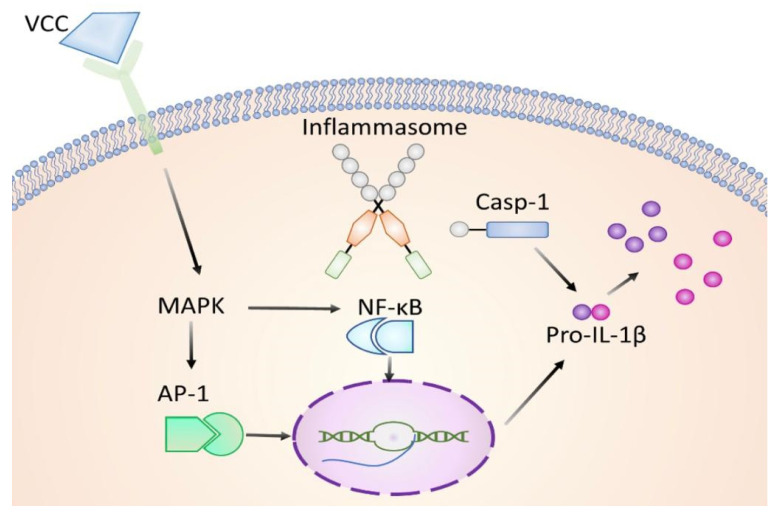
Model mechanism proposed for the monomeric VCC starting the innate immune response. The monomeric form of the VCC behaves as a PAMP; it binds its membrane receptor (TLR 4), turning on a signaling cascade activating MAPKs (ERK, p38 and perhaps JNK). The MAPKs then start the activation of the transcription factors AP-1 and NF-κB, starting the pro-inflammatory and innate response involved with the synthesis of the classic IL-1β.

**Figure 9 ijms-24-07272-f009:**
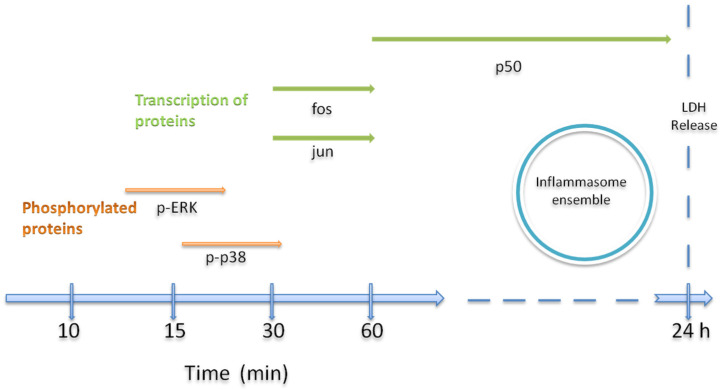
Proposed time lapse of activated MAPKs and their transcriptional activators. Phosphorylated MAPK proteins and activated transcripts of AP-1 (Fos, Jun) and NF-κB elicited by treatments with the monomeric VCC on THP-1-treated cells.

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
