# Peer review of "Transcriptional Activation of a Pro-Inflammatory Response (NF-κB, AP-1, IL-1β) by the Vibrio cholerae Cytotoxin (VCC) Monomer through the MAPK Signaling Pathway in the THP-1 Human Macrophage Cell Line"

_ijms, 2023, doi:10.3390/ijms24087272_

Round 1
Reviewer 1 Report
The authors showed that the monomeric soluble form of VPC modulates the innate immune response in macrophages, which is consistent with NLRP3 inflammasome assembly actively releasing IL-1b, and also induces pyroptosis controlled by LDH release.
Please, provide original pictures for western-blots.
What thoughts do the authors have about why the Pro IL-1b band in Fig. 1a has a white center?
The authors showed that 40 ng/ml is a nonvacuolating concentration. It is unclear why the authors used a non-vacuolizing concentration of 40 or 42 pg/ml in the following experiments? Is it typo?
The authors need to present the results of a complete experiment with dilutions of LPS to determine the maximum concentration of LPS that does not cause THP-1 cell vacuolization.
The authors need to quantify the Western blotting results in Figure 1B, 7B
Please provide more information about VCC purification and determination of LPS contamination in protein preparations? Have you used additional steps to remove LPS from protein preparations?
Because the authors tested the experiments on THP-1, which is a cancerous line that has various cellular processes disrupted and can be expected to react abnormally compared to normal macrophages, they need to describe the limitations of their work at the end of the Discussion section.
In the abstract, please check the meaning of the sentence "Nevertheless, the knowledge of this mechanism in the human immune system is still perfectible." Is there a typo in the last word? Please also check for the correct spelling of words throughout the manuscript.
Please enter the abbreviations LPS, PMA, and ATP. Please check other abbreviations throughout the text as well.
Figure 7a. The authors need to add LDH activity data for LPS at the 24-hour time point. What does C- stand for? Please clarify this in the caption of Figure 7a.
In figures 1b and 7b, please use the English form of the word "b-actina".
In the caption of Figure 7b, please use the English form of the word "Caspasa 1".
Author Response
Reviewer 1
The authors showed that the monomeric soluble form of VPC (VCC) modulates the innate immune response in macrophages, which is consistent with NLRP3 inflammasome assembly actively releasing IL-1b, and also induces pyroptosis controlled by LDH release.
- Please, provide original pictures for western-blots.
Original pictures of the WB of this study are provided as a series of figures (Supplement 1).
- What thoughts do the authors have about why the Pro IL-1b band in Fig. 1a has a white center?
A white center of the band is due to over-exposure to the densitometer, we apologize for it. Other experiments are shown in Supplement 1.
- The authors showed that 40 ng/ml is a nonvacuolating concentration. It is unclear why the authors used a non-vacuolizing concentration of 40 or 42 pg/ml in the following experiments? Is it typo?
40 ng/ml was an error of typing, it should say 40 pg/ml and it was already changed.
Also whenever there was 42 pg/ml it should be 40 pg/ml, it was an error by the paper transcription.
A non vacuolating concentration was selected by titration of the toxin because we were interested in a cell response due to membrane receptors, such as TLRs (present in the THP-1 cell line). It is established by the literature, the use of LPS as the major inducer to study the inflammasome in THP-1 cell line.
- The authors need to present the results of a complete experiment with dilutions of LPS to determine the maximum concentration of LPS that does not cause THP-1 cell vacuolization.There’s a misunderstanding in the phrasing, since LPS does not cause vacuolization. VCC being a pore forming toxin, causes vacuolization due to autophagy events following internalization of membrane containing the heptameric form of the toxin, that usually causes pores, punching the cytoplasmic membrane. The membrane is repaired by the cell performing endocytosis and it is observed at concentrations of the order of nanograms. The endocytosis at such concentrations of the monomer elicit so many events at the same time, that autophagy is triggered (Figueroa-Arredondo et al. 2001).
- Concentrations in the order of picograms were used, since the vacuolating effect is avoided due to the lower availability of the toxin monomers. Nevertheless, picogram concentrations are enough to raise a response, as it is shown through the figures. The experiment of different concentrations of the toxin is included as the Supplement No. 2.
- The authors need to quantify the Western blotting results in Figure 1B, 7B.
We did not quantify results of the western blots, since they are not experiments of kinetics or dose/response. WBs are not planned to show quantification, only to show presence or lack of response.
- Please provide more information about VCC purification and determination of LPS contamination in protein preparations? Have you used additional steps to remove LPS from protein preparations? VCC purification is performed by molecular weight exclusion membranes; later gel electrophoresis PAGE was performed and then the protein recovered by overlapping the band of 65 kDa in the Western blot, the band was eluted and recovered from the gel. Identification of the monomer was done, using a polyclonal antibody towards the toxin. The procedure excludes high molecular weight molecules such as LPS. Accidental contamination was carefully avoided during the whole process, yet the cell line is responsive only to nanograms of LPS, which would be practically impossible to achieve under the present conditions, since to reach the picograms per ml concentration used in this study, the stock of the toxin was diluted 3000 times. (Supplementary Information 2). Additionally, in experiments where LPS is added, sonication is required to make available the highly insoluble material, the protein in these experiments was never sonicated.
- Because the authors tested the experiments on THP-1, which is a cancerous line that has various cellular processes disrupted and can be expected to react abnormally compared to normal macrophages, they need to describe the limitations of their work at the end of the Discussion section. Although the THP-1 cell line is derived from acute myeloid leukemia, it shows membrane markers indicating that it still belongs to the lineage of granulocyte monocyte progenitors (Adati et al 2009, Liu et al 2019). THP-1 has more attributes of monocytes in comparison to other human myeloid cell lines. The cell line has preserved the TLR-4/ NFkB pathway and that makes it suitable to perform the studies characterizing the LPS response (Sharif et al. 2007); and that is the reason that THP-1 cell line was chosen as the right system to perform this study. Being a cell line derived from cancer, several other signaling cascades that are not functional and cannot be studied using them. Then, to the purposes of the present work, it is a suitable cell line, since we want to establish a response involving the pathway of TLR-4. The present work shows experimental results on a cell line mimicking peripheral blood monocytes, with active inflammatory response pathways (Anderson and Sundler 2000). A paragraph note underlying the above, will be included in the Discussion section as suggested.
- In the abstract, please check the meaning of the sentence "Nevertheless, the knowledge of this mechanism in the human immune system is still perfectible." Is there a typo in the last word? A mistake was made in the choice of the word “perfectible” and it will be eliminated. The sentence will be changed by the following: “Nevertheless the knowledge of the pathway of human macrophages in response to the monomer of the VCC, is still susceptible of further description¨.
- Please also check for the correct spelling of words throughout the manuscript.
The spelling was checked in the version here submitted.
- Please enter the abbreviations LPS, PMA, and ATP. Please check other abbreviations throughout the text as well.
Abreviations of LPS, PMA, and ATP were included.
Abbreviations were reviewed throughout the MS as requested.
- Figure 7a. The authors need to add LDH activity data for LPS at the 24-hour time point
It was considered unnecessary to show LDH released by the LPS controls, since it has been shown by authors in the literature already, please see the requested information in Supplementary information 2.
- What does C- stand for? Please clarify this in the caption of Figure 7a.
A mistake was made by mentioning C- instead of Without Treatment Control (W/T). The error was corrected in the present version of the MS.
- In figures 1b and 7b, please use the English form of the word "b-actina".
This error was corrected in the present version of the MS.
- In the caption of Figure 7b, please use the English form of the word "Caspasa 1".
This error was corrected in the present version of the MS.
Bibliography.
Adati, N., Huang, M.-C., Suzuki, T., Suzuki, H., Kojima, T., 2009. High-resolution analysis of aberrant regions in autosomal chromosomes in human leukemia THP-1 cell line. BMC Res. Notes 2, 153.
Andersson, K., Sundler, R., 2000. Signalling to translational activation of tumour necrosis factor-α expression in human THP-1 cells. Cytokine 12, 1784–1787.
Liu, Z., et al., 2019. Fate mapping via Ms4a3-expression history traces monocyte-derived cells. Cell 178, 1509–1525.e19.
Sharif, O., Bolshakov, V.N., Raines, S., Newham, P., Perkins, N.D., 2007. Transcriptional profiling of the LPS induced NF-κB response in macrophages. BMC Immunol. 8, 1.

Reviewer 2 Report
There should be no links in the conclusion, move it to the discussion section. The list of literature needs to be significantly increased. Please note that the article is dated 21 years, but the editors will probably correct it to 23. In the Acknowledgments section, not quite accepted forms are written, this is a scientific journal, not an inscription on the fence, so "We love you" is not entirely appropriate. The affiliation is written in Spanish. Rewrite the conclusion. The paragraph that begins with the words of the Colleague .... move it to the discussion
Author Response
Suggestions and comments Reviewer 2.
- There should be no links in the conclusion, move it to the discussion section.
No links were kept in the conclusion of the updated version.
- The list of literature needs to be significantly increased.
The list of references will be increased as requested in an updated version of the MS. Is currently in progress.
- Please note that the article is dated 21 year, but the editors will probably correct it to 23.
Observation was noticed, it will be mentioned to the editor in written by email.
- In the Acknowledgments section, not quite accepted forms are written, this is a scientific journal, not an inscription on the fence, so "We love you" is not entirely appropriate.
Acknowledgments section rewritten.
- The affiliation is written in Spanish.
I apologize and kindly request the affiliation to be kept. This is because it is a requirement from the mentioned institutions that their names are in Spanish, since that is the only way those Institutions can be found in databases.
- Rewrite the conclusion.
The paragraph that begins with the words of the Colleague .... move it to the discussion.
Conclusion rewritten.
Thank you for your help.

Round 2
Reviewer 1 Report
The authors have made significant edits to the manuscript in accordance with the reviewers' comments. I have no further comments on the revised version of the manuscript. When submitting the paper, please mark the Supplementary Material correctly so that it is not in the form of a cover letter.